# Recrudescence of transmission of onchocerciasis in some endemic communities in Kaduna State, Nigeria: What is the way forward?

Timothy O. Olanrewaju[1]*, Felicia N. C. Enwezor[1], Luret A. Lar[2], Michael A. Igbe[3], Ramatu A. Abdullahi[1], Monsuru A. Adeleke[4], Oluwatosin B. Adekeye[5], Elizabeth O. Elhassan[6]

**1** Nigerian Institute for Trypanosomiasis (*and Onchocerciasis*) Research, Kaduna, Kaduna State, Nigeria, **2** Department of Community Medicine, Faculty of Clinical Sciences, College of Health Sciences, University of Jos, Jos, Plateau State, Nigeria, **3** Federal Ministry of Health, Federal Capital Territory, Abuja, Nigeria, **4** Department of Zoology, Faculty of Basic and Applied Sciences, Osun State University, Osogbo, Osun State, Nigeria, **5** Department of Psychiatry, Ahmadu Bello University Teaching Hospital Shika, Zaria, Kaduna State, Nigeria, **6** Freelance Consultant, Ungwan Rimi, Kaduna State, Nigeria

\* timothyseye@gmail.com

## Abstract

### Background

Onchocerciasis caused by the filarial parasite *Onchocerca volvulus* and transmitted by *Simulium damnosum* s.l. remains a public health concern in Nigeria. Infestation of *S. damnosum s.l.* along rivers Gurara and Kaduna; and heavy intensity of *O. volvulus* infection in Kaduna were documented in 1956. Control of onchocerciasis in Kaduna started 1954 with larviciding using dichlorodiphenyltrichloroethane reduced *S. damnosum s.l.* population by 94% in 1966. Diethylcarbamazine used for human treatment was discontinued due to toxicity. Kaduna State Ministry of Health and its partners built on this achievement; used annual mass administration of ivermectin consistently between 1989 and 2017 which led to interruption of onchocerciasis transmission in 2018. This study investigated possibility of recrudescence of onchocerciasis with the hypothesis that insecurity-induced migration could cause recrudescence of onchocerciasis.

### Methodology/Principal findings

Six out of the 23 Local Government Areas (LGAs) in Kaduna State were selected for evaluation. Adult *S. damnosum* s.l. were captured across seven breeding sites using human landing collectors from July to October 2023. Pooled screen assays of 72 pools heads of black flies were conducted using quantitative polymerase chain reaction (qPCR) for *O. volvulus* detection. Dried blood samples from 3107 children aged 5–9 years were collected; with 1502 samples analysed using alkaline phosphatase enzyme-linked immunosorbent assay due to financial constraints. Twenty pools of the

**Data availability statement:** All relevant data are within the manuscript and its Supporting Information files.

**Funding:** This work received financial support from the Coalition for Operation Research on Neglected Tropical Diseases (COR-NTD), which is funded at The Task Force for Global Health primarily by the Bill and Melinda Gates Foundation and the United States Agency for International Development through its Neglected Tropical Diseases Program. The grant was administered by the African Research Network for Neglected Tropical Diseases under Small Grant Program-VI (SGP159/050.2023) to TOO. The funders had no role in study design, data collection and analysis, decision to publish, or preparation of the manuscript. https://www.gatesfoundation.org/; https://www.usaid.gov/

**Competing interests:** The authors have declared that no completing interests exist.

72 pools of heads (27.8%) of *S. damnosum s.l.* analysed were positive for *O. volvulus* in Kagarko and Kachia LGAs (> 1/2000 infective flies; 95% upper confidence limit 0.49) with qPCR prevalence of 0.32%. Two children from security compromised communities tested seropositive (prevalence 0.31%; 95% upper confidence limit 0.317).

## Conclusion/Significance

The findings demonstrated ongoing onchocerciasis transmission in Kaduna despite the interruption in 2018. This calls for evaluation of the extent of recrudescence and identification of key drivers such as human migration, fly movement and insecurity.

## Author summary

Onchocerciasis is a disease that affects humans and is transmitted by bites of *Onchocerca volvulus* infected black flies. Nigeria accounts for the highest onchocercal blindness in sub-Saharan Africa. Collaborative effort by various stakeholders in Kaduna State, Nigeria yielded remarkable achievement of successful interruption of onchocerciasis transmission in year 2018 through annual mass administration of medicine (MAM). However, our study established the ongoing transmission of onchocerciasis in some areas of Kaduna State five years after a stop-MAM decision. Kaduna State Government, partners and stakeholders should urgently take actions to mitigate the ongoing transmission of onchocerciasis in the State to safeguard the achievements attained after years of MAM. We recommend the following; assessment of entire Kaduna State to determine the extent of the spread of the recrudescence; in-depth analysis of annual therapeutic coverage rates; community-driven slash and clear approach for vector control; review of the monitoring and evaluation system across all levels; using Esperanza Window Trap at black flies breeding sites to complement the slash and clear approach; modelling the approaches to determine the most appropriate frequency of treatment with ivermectin.

## Introduction

Human infection with the filarial parasite *Onchocerca volvulus* causes onchocerciasis (also known as river blindness). *Onchocerca volvulus* is transmitted to humans by bites of *Simulium damnosum s.l.* (black flies) in Nigeria. There are nine sibling species within the *S. damnosum* complex that can transmit onchocerciasis [1,2]. Onchocerciasis is one of the Neglected Tropical Diseases (NTDs) with high public health impact in Africa. Over 99% of all cases of onchocerciasis and onchocercal-related blindness are found in Africa; with Nigeria having about 40% of the global population at risk [3]. Onchocercal blindness ranked second leading cause of preventable blindness in the world after trachoma [4]. The disease is peculiar to the poor with economic implications resulting in abandonment of rich fertile farmlands which resulted in food insecurity and poor school attendance, sometimes forcing children to drop out

of school to assist blind parents/guardians [5]. Other manifestations include skin disfiguration and stigmatization. These impacts on the wellbeing of individuals with the disease become a major challenge to the achievement of Sustainable Development Goals (SDGs); "End hunger, achieve food security and improved nutrition and promote sustainable agriculture"(SDG 2) and "Ensure healthy lives and promote well-being for all at all ages (SDG 3) as documented by Engels [6].

Crosskey [7] documented that adult *S. damnosum s.l.* dispersal ability is estimated to be 234,000 $Km^2$ (90,000 square miles) in Northern Nigeria. Nearly half of the areas infested by *S. damnosum s.l.* in Northern Nigeria lies within region of rivers Gurara and Kaduna which is about 37,400 $Km^2$ (14,400 square miles). In Nigeria, a State is classified as a transmission zone [8]. The control of onchocerciasis in Kaduna State started in 1954 with larviciding using dichlorodiphenyltrichloroethane (DDT) along rivers Gurara and Kaduna [9,10]. By 1966, this control effort had achieved 94% reduction in *S. damnosum s.l.* population density. Diethylcarbamazine was also used for human treatment but discontinued due to high toxicity. The State started annual mass administration of medicine (MAM) of ivermectin in 1989. Tekle et al [11] reported that 17 years of consistent annual administration of ivermectin could potentially interrupt transmission of onchocerciasis in two foci (Birnin Gwari and Kauru/Lere) within Kaduna State. By 2018, interruption of transmission of the disease was attained and MAM implementation was stopped [12].

However, the current security challenges which revolve around ethno-religious and farmer-herder conflicts over land access heighten violence in Kaduna State [13]; migration to the State from the border States like Niger, Kano and Federal Capital Tertiary (FCT) Abuja where transmission of onchocerciasis is still ongoing [14] and vector migration due to climate change could cause recrudescence of onchocerciasis in Kaduna State. According to the International Organization for Migration (IOM) report [15], Kaduna State has a minimum of 117,880 internally displaced persons (IDPs) as a result of banditry and kidnapping (45%), communal clashes (29%), natural disaster (13%), farmers-herders clashes (12%), and insurgency (2%). Also, it is imperative to state that 38,900 IDPs (33%) of the total IDPs identified in Kaduna by IOM in year 2023 round 12 assessment migrated to Kaduna State from other neighbouring States. Our interaction with the State Emergency Management Agency (SEMA) revealed that the only State government managed IDP camp is located at Giwa Local Government Area (LGA). However, we identified a faith-based IDP camp at Agwan Zawu, Gonin-Gora, Chikun LGA and another at Evangelical Church Winning All (ECWA), Ungwan Musa, Zonkwa, Zangon-Kataf LGA. Interruption of transmission of onchocerciasis in Kaduna State might be difficult to sustain because of high number of IDPs within the State without any organized or governmental camp. Also, increase in inter-state migration within northwest region of Nigeria and climate change could affect transmission of onchocerciasis. These challenges could jeopardise efforts towards 2030 target of elimination of onchocerciasis, other targets in the NTD Roadmap 2021–2023 and SDG 2 and 3.

We investigated possible recrudescence of onchocerciasis transmission in six LGAs in Kaduna State in the face of insecurity and human migration five years after interruption of transmission.

## Methods

### Ethics statement

The study protocol was approved by the Health Research Ethics Committee (HREC) of Kaduna State Ministry of Health (Ref No. NHREC/17/03/2018). Verbal assent was obtained from each child with verbal consent of either parent or guardian of the child before blood sample collection. Volunteers that acted as human attractants for black fly catching were told about the personal risks and community benefits of participation and given the option to opt out of participation at any time without any repercussions after signing the participant consent form. They were offered ivermectin at the end of the study according to specifications.

### Study area

Kaduna State lies between 10°36'33" N and 7°25'46" E and shares boundaries with Kano and Katsina States in the north; Plateau State to the east; Nasarawa State to the south; Niger State in the west and the FCT Abuja to the southwest.

Neglected Tropical Diseases

PLOS

Some of the major rivers in the State include river Gurara and river Kaduna which serve as good breeding sites for black flies [16]. All of the States that share boundaries with Kaduna are currently treating with ivermectin except Plateau and Nasarawa States, which have interrupted transmission, and Katsina State which was hypo endemic for onchocerciasis and was ineligible for treatment requires onchocerciasis elimination mapping (OEM). Kaduna State consists of 23 Local Government Areas (LGAs) with a projected population of more than six million people and estimated land area of about 46,063 $Km^2$. The demographic vegetation of the State is mainly savannah grassland with few areas of forest. This study was carried out in six LGAs of Kaduna State, namely Kaduna North, Kaduna South, Chikun (Fig 1), Kachia, Kagarko and Zangon-Kataf (Fig 2). Kaduna North and Kaduna South LGAs were selected due to the presence of black fly breeding sites at river Kaduna that runs through the two LGAs and influx of migrants from Chikun LGA. Chikun, Kachia and Kagarko LGAs were included because of the presence of black flies breeding sites along rivers Gurara and Kaduna; insecurity and presence of migrants. Zangon-Kataf LGA was chosen based on high level of insecurity and proximity to breeding site in Kachia. Seven breeding sites across five LGAs served as catching points of black flies namely Kabala Doki (Kaduna North LGA), Kwata (Kaduna South LGA), Tsallake (Chikun LGA), Atara and Dogon-Daji (Kagarko LGA), Amuse and Gurara (Kachia LGA) as shown in Table 1.

## Epidemiological evaluation

The epidemiological study was carried out in all the selected LGAs. Sample collection was done between July and October, 2023.

**Study population.** Farming is the major occupation of citizens at Chikun, Kagarko, Kachia and Zangon-Kataf, but for Kaduna South and Kaduna North LGAs majority are civil servants and business men/women. Dried blood samples were collected from children (male and female) between 5 and 9 years old across the study areas.

**Sample size and sample collection.** Dried blood samples (DBS) were collected from 3107 children aged 5–9 years from various communities and primary schools across the six LGAs using World Health Organization (WHO) guidelines for epidemiological assessment of onchocerciasis transmission in children [8,17]. The sample from each community was estimated proportionally to available children within the required age group. Blood samples were collected from all assented children aged 5–9 years through finger-prick using sterile lancet after swabbing the finger with methylated spirit; a lancet to each child. About 100 µL of blood from each child was spotted on Whatman number 1 filter paper (Whatman International Ltd Maldstone, England); using one filter paper for a child. Each of the filter papers was fixed to a Styrofoam using pencil for air-drying. For each community, each child's sample was stored in a small ziplock bag and then put into a larger ziplock bag containing silica gel (Desicare, Reno, NV, USA), labelled appropriately and stored at 25ºC.

**Sample analysis.** The samples were analysed within six months of sample collection at Osun State University Multi-Disciplinary Research Laboratory, Osogbo, Nigeria using AP-ELISA technique as detailed in S1 Methods.

## Entomological evaluation

Seven known breeding sites of *S. damnosum s.l.* in five LGAs – Chikun, Kachia, Kaduna North, Kaduna South and Kagarko were assessed to ascertain if they were currently breeding the vector. The coordinates of the breeding sites were taken using Garmin geographical positioning information system (GPIS) machine. Based on the outcome of the assessments, community leaders of first-line communities (communities within 20km to *S. damnosum s.l.* breeding site) and second-line communities (communities within 20km – 40km away *S. damnosum s.l.* breeding site) were sensitised on the importance of preventing blindness due to onchocerciasis. The leaders identified two persons from their communities that serve as human landing collectors (HLCs) for *S. damnosum s.l.* The HLCs were clustered into two groups for a two day theory and practical training on capturing of black flies and recording as described [18,19]. Black fly capturing activities were conducted between July and October 2023 being the peak of black fly breeding season. Two HLCs worked in an

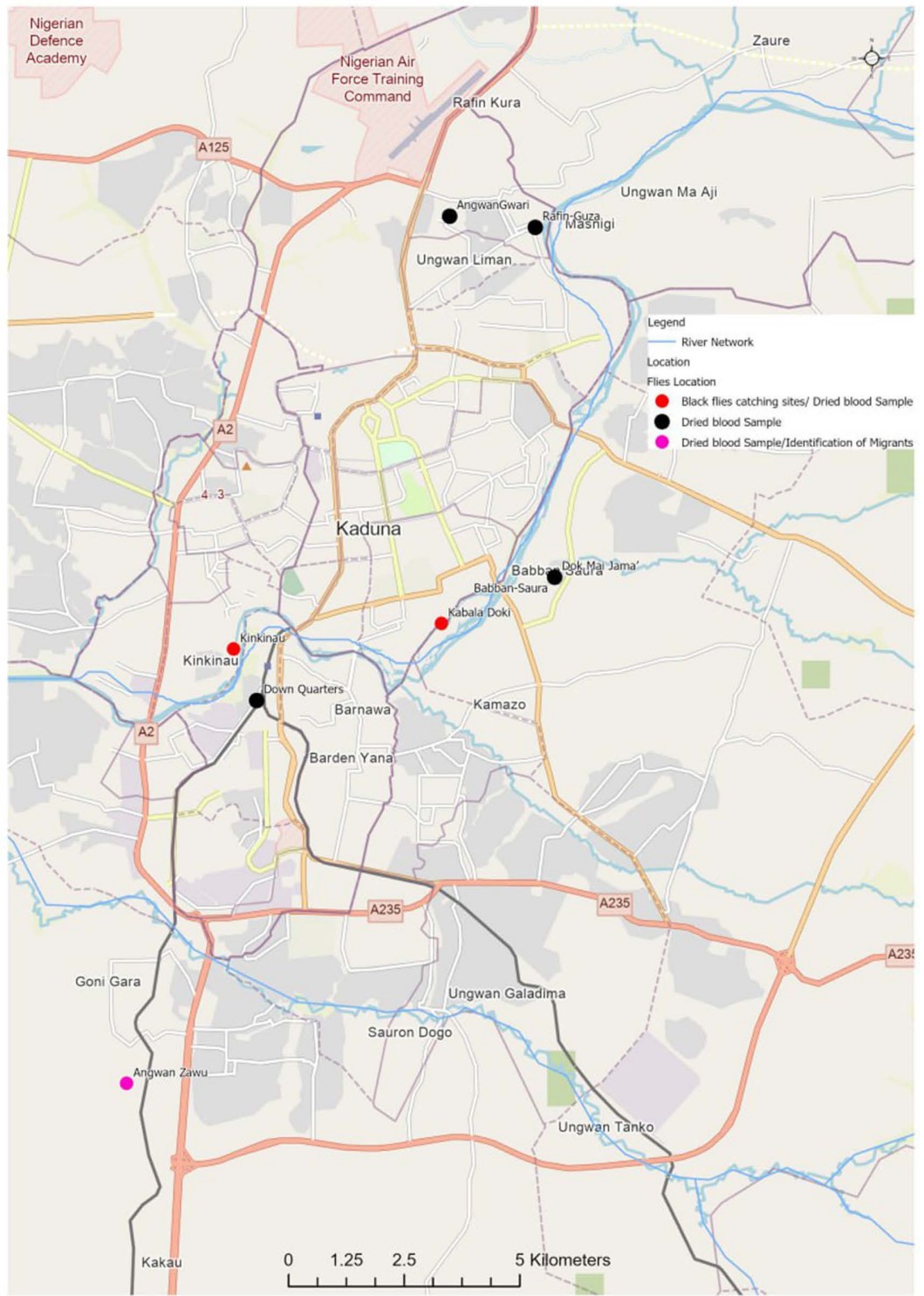

**Fig 1. Map of Kaduna State showing the study locations; Kaduna North, Kaduna South and Chikun LGAs (ArcGIS Pro Software version 3.2.0;** https://qgiscloud.com/Ayosam/Kaduna_Metropolis_Project_Area/?l=River%20Network%2CStudy%20Area%2CClassification%20of%20Survey%20Site&bl=Road&t=Kaduna_Metropolis_Project_Area&e=511398%2C930172%2C1234240%2C1265134**).**

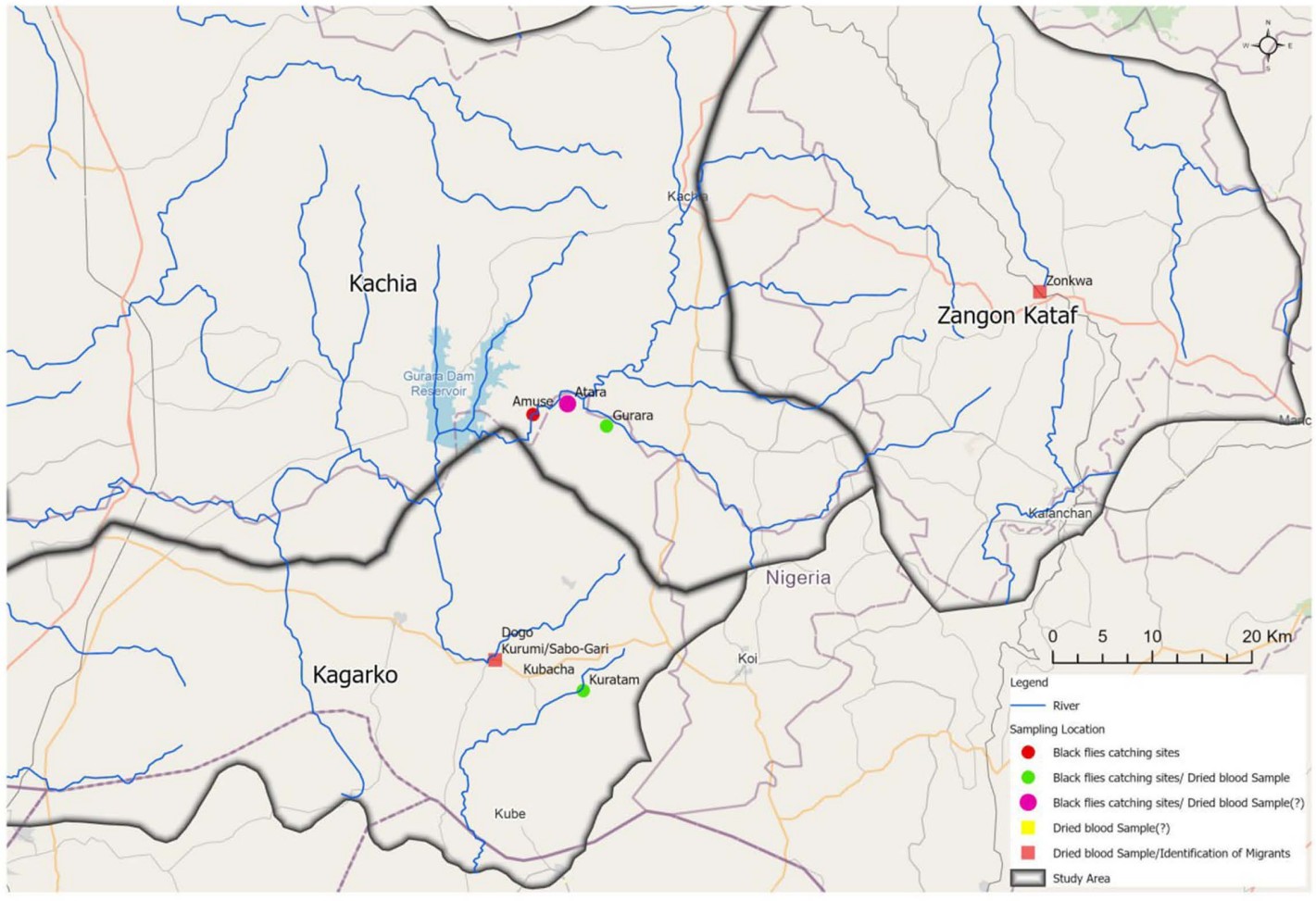

**Fig 2. Map of Kaduna State showing the study locations; Kachia, Kagarko and Zango-Kataf LGAs (ArcGIS Pro Software version 3.2.0; https://qgiscloud. com/Ayosam/Map_2/?l=Sampling%20Location%2CRiver%2CStudy%20Area&bl=mapnik&t=Map_2&e=521670%2C910793%2C1244512%2C1245755).**

alternate arrangement hourly for two days per week between 07:00 h and 18:00 h. The flies collected were stored in preservative bottles containing isopropyl alcohol. The bottles were labelled accordingly for each catching site. At the end of capturing of the flies, the study entomologist sorted all the flies and confirmed the flies that were *S. damnosum s.l.* using morphological characteristics such as conspicuous compound eyes, short horn-like antennae, hump-backed at midsection of the thorax and broad, colourless and transparent wings. The black flies were stored in isopropyl alcohol and hand carried to the Multi-disciplinary Research Laboratory, Osogbo, Nigeria to determine the *O. volvulus* infectivity rate in black flies using real-time qPCR technique assay with a mitochondrial DNA target as detailed in S2 Methods. The separation of the heads from the bodies, extraction and qPCR were carried out as described [20].

## Statistical analysis

The number of black flies collected in each location was normalized and then subjected to One -Way Analysis of Variance using SPSS version 23.0 to determine the significant difference in fly variation across catching sites and also the months

**Table 1. Dried blood sample Ov16 antibody/antigen test using AP-enzyme linked immunosorbent assay.**

| Local government area | Community | Proximity to black fly breeding site | Number of children sampled | No. of DBS analysed | No. of positive DBS |
|---|---|---|---|---|---|
| Kaduna North | Kabala | First line | 304 | 86 | 0 |
| | Angwan-Gwari | Second line | 136 | 136 | 0 |
| | Rafin-Guza | Second line | 160 | 100 | 0 |
| | Kawo | Second line | 106 | 0 | 0 |
| Kaduna South | Kinkunu | First line | 176 | 100 | 0 |
| | Sheilk Gumi Primary School | Second line | 233 | 0 | 0 |
| | Down Quarters | Second line | 136 | 135 | 0 |
| | Television | Second line | 134 | 0 | 0 |
| | Aliyu Makarma Primary School | Second line | 276 | 0 | 0 |
| Chikun | AngwanZawu IDP camp | Second line | 23 | 23 | 0 |
| | Doka Mai Jama' | First line | 137 | 137 | 0 |
| | UBE Bagado Primary School | Second line | 122 | 0 | 0 |
| | Babban Saura | First line | 237 | 120 | 0 |
| Kagarko | Kuratam | First line | 231 | 231 | 0 |
| | Atara | First line | 76 | 76 | 1 |
| | Kubacha | Second line | 100 | 100 | 1 |
| | Dogo Kurumi | Second line | 71 | 71 | 0 |
| Kachia | Gurara | First line | 49 | 49 | 0 |
| | Zonkwa Primary School | Second line | 262 | 0 | 0 |
| Zangon-Kataf | Samaru Kataf | Second line | 138 | 138 | 0 |
| TOTAL | | | 3107 | 1,502 | 2 |
| | Lower and Upper Limit CI (95%) | | | | 0.13 (-0.051 - 0.368) |

The 95% UCL of OV-16 positivity > 0.1% indicates ongoing transmission of Onchocerciasis in Kaduna State.

of collection as detailed in S3 Methods. The Ov16 antibody prevalence in children was determined using 95% confidence intervals of > 0.1% upper confidence limit (UCL) as described [21] using the formula:

95% Confidence interval = proportion positive (p) ± 1.96 $\left( \frac{\sqrt{P(1-P)}}{n} \right)$. The poolscreen software 2.1 was used to

determine *O. volvulus* infectivity rate in pools of black flies at 95% confidence interval.

## Results

Out of the 1,502 DBS of children aged 5–9 years; two children from Kagarko LGA were Ov16 sero-positive with prevalence of 0.13% (95% upper confidence limit 0.317) as shown in Table 1. These children were from first-line and second-line communities, black fly breeding sites; and communities with IDPs and migrants. The two Ov16 sero-positive samples were from Atara community (first-line community) and Kubacha community (second-line community) within Kagarko LGA. Six thousand eight hundred and ninety-seven flies out of 7,782 reported flies from the field were identified as *S. damnosum s.l.* (Table 2). The highest number 2,858 (41.44%) of black flies captured was from Kuratam community followed by 2,072 (30.04%) in Atara within Kagarko LGA. Gurara and Amuse communities had 1,181 (17.12%) and 782 (11.34%) respectively in Kachia LGA. Only four black flies were captured along river Kaduna; two at Kabala and two at Kinkinau. We observed a significant variation in the number of flies captured at various catching sites during the wet

**Table 2. Human landing capturing of flies at various breeding sites in the study areas.**

| Local Government Area | Community | Catching Site | Number of fly catchers | Number of flies captured in the field | Number of con-firmed black flies in the laboratory | Number of pool | Black fly popula-tion density in wet season (%) | Mean+SD (*p<0.05*) |
|---|---|---|---|---|---|---|---|---|
| **Kaduna North** | Kabala | Kabala | 2 | 148 | 2 | 1 | 1.4 | 37.00+8.25[a] |
| **Kaduna South** | Kinkinau | Kwata | 2 | 27 | 2 | 1 | 7.4 | 6.75+3.86[a] |
| **Chikun** | Babban Saura | Tsallake | 2 | 11 | 0 | 0 | 0 | 2.75+1.89[a] |
| **Kagarko** | Atara | Atara | 2 | 2233 | 2072 | 21 | 92.8 | 558.25+148.30[b] |
| | Kuratam | Dogon Daji | 2 | 2924 | 2858 | 29 | 97.7 | 731.00+70.77[c] |
| **Kachia** | Amuse | Gurara | 2 | 1175 | 782 | 8 | 66.6 | 293.75+113.72[d] |
| | Gurara | Gurara | 2 | 1264 | 1181 | 12 | 93.4 | 316.00+8.36[d] |
| **TOTAL** | | | **7** | **14** | **7,782** | **6,897** | **72** | |

The mean±SD (standard deviation) with different superscript along the same column indicates statistically significant difference (*p<0.05*).

season as the number of black flies collected at Kuratam, Atara, Gurara and Amuse were significantly higher (*p>0.05*) than those collected in Kabala, Kinkinau and Babban Saura (Table 2). There was significant variation (*p>0.05*) in the number of flies collected during the four months catching period (Table 3).

Twenty pools of the 72 pools of heads (27.8%) of *S. damnosum s.l.* analysed were positive for *O. volvulus* (> 1/2000 infective flies; 95% upper confidence limit 0.49 of black flies' infectivity) with qPCR prevalence of 0.32% (Table 4). Four out of the seven black flies catching sites (57.14%) tested positive with both sites in Kagarko and Kachia LGAs testing positive. We observed the highest infectivity rate (33%) in black flies captured at Amuse catching site in Kachia LGA.

## Discussion

Black flies prefer breeding in conducive river systems where the physico-chemical parameters of the rivers support their establishment [9]. The results of the present study showed that the river system in Kaduna indeed support the breeding of the black flies, though with variation as evident in the current study. Historical data showed that Davies et al [9] carried out larviciding using DDT along river Gurara from Suleja, Abuja to Tafa due to the high bit-ing density of black flies along the river. This resulted to 94% reduction in the population density of *S. damnosum* in the area. On the one hand, the high number of flies collected along river Gurara suggests that breeding of black fly continues along the river. On the other hand, the high water level in river Kaduna resulted in low density of black fly in Chikun, Kaduna North and Kaduna South LGAs despite four months attempt to capture black flies. The low density of black flies could be attributed to flooding, inadequate rocky basements and low velocity of water. According to Opara et al [22] and Eyo et al [23], heavy rainfall and stormy weather played major role in washing away most breeding sites of black flies during the rainy season. Inadequate rapids and rocky basements which form favourable breeding sites for black flies [24] might also affect the population density. Furthermore, high water level during heavy rain can slow down the velocity of water which reduces the favourable conditions for breeding of *Simulium* species [25].

Though the breeding sites along river Gurara were probably not visited during cytotaxonomic identification of members of *S. damnosum* complex [26], breeding of black flies and transmission of onchocerciasis along the river at Gantang in Kagarko LGA has been documented [16]. The high population density of black flies recorded in Amuse, Gurara, Atara and Kuratam sites could be because of vegetation, rocks, speed of the water and other favourable physico-chemical

Neglected Tropical Diseases

**Table 3. Variation in fly population density during wet season.**

| Local Government Area | Community | July | August | September | October |
|---|---|---|---|---|---|
| Kaduna North | Kabala Doki | 30 | 42 | 46 | 30 |
| Kaduna South | Kinkinau | 12 | 7 | 5 | 3 |
| Chikun | Babban Saura | 0 | 4 | 3 | 4 |
| Kagarko | Atara | 587 | 619 | 683 | 344 |
| | Kuratam | 650 | 760 | 813 | 701 |
| Kachia | Amuse | 387 | 349 | 309 | 130 |
| | Gurara | 318 | 333 | 293 | 320 |
| Total | | 1984 (25.5%) | 2114 (27.2%) | 2152 (27.7%) | 1532 (19.70%) |

**Table 4. Detection of *Onchocerca volvulus* DNA signal in *S. damnosum s.l.* pools using qPCR.**

| Local government area | Community | No. of black flies analysed | No. of pools of 100 | No. of pools < 100 | No. of positive pools | Prevalence by Ov ND5 qPCR (95% confidence interval) |
|---|---|---|---|---|---|---|
| Kaduna North | Kabala Doki | 2 | 0 | 1(2) | 0 | 0.00 (0.00) |
| Kaduna South | Kinkinau | 2 | 0 | 1(2) | 0 | 0.00 (0.00) |
| Chikun | Tsalake | 0 | 0 | 0 | 0 | 0.00 (0.00) |
| Kagarko | Kuratam | 2,858 | 28 | 1(58) | 7(24%) | 0.0275 (0.13-0.545) |
| | Atara | 2,072 | 20 | 1(72) | 7(33%) | 0.044 (0.19 – 0.79) |
| Kachia | Gurara | 1,181 | 11 | 1(81) | 3(25%) | 0.0287 (0.09-0.76) |
| | Amuse | 782 | 7 | 1(82) | 3(38%) | 0.047 (0.15-1.18) |
| TOTAL | | 6,897 | 66 | 6 | 20 | 0.32 (0.21-0.49) |

The 95% UCL of black flies infectivity of >1/2,000 infective flies/2,000 indicates ongoing transmission. CT-values for pools of black flies are detailed in S1 Results.

parameters of the river Gurara [27,28]. The population density along river Gurara corroborates the earlier report [16]. The high infectivity rate in black flies we observed in Amuse, Atara, Gurara and Kuratam suggests that transmission cycle of onchocerciasis has been sustained for a while in Kachia and Kagarko LGAs.

Earlier studies by [7] identified five transmission foci along the rivers in Kaduna State (Zaria, Gurara, Kaduna Central, Birnin Gwari and Kauru/Lere). The present study covered only two transmission foci (Gurara and Kaduna Central) out of five transmission foci in Kaduna State. Tekel et al [11] evaluated impact of continuous treatment with ivermectin on transmission of onchocerciasis in two transmission foci of Birnin Gwari and Kauru/Lere and reported zero prevalence of *O. volvulus* in 3,703 skin-snipped individuals from 27 communities evaluated after 15 – 17 years continuous treatment with ivermectin. Furthermore, Isiayaku et al [12] carried out epidemiological and entomological assessments of onchocerciasis for stop-MAM assessment in all the five transmission foci in Kaduna State within 2016 and 2018 after 20 years of consistent treatment with ivermectin in accordance to WHO 2016 onchocerciasis elimination guidelines. While Tekel et al [11] alluded to the possibility of elimination of onchocerciasis in Kaduna State after consistent treatment with ivermectin, Isiayaku et al [12] categorically documented attainment of interruption of transmission of onchocerciasis in Kaduna State after prolonged administration of ivermectin.

After five years post interruption of transmission of onchocerciasis in Kaduna State, we investigated seven black flies breeding sites of which two (Gurara with Latitude 9.665326; Longitude 7.883456 and Kuratam with Latitude 9.600633; Longitude 7.816031) were among the sites earlier investigated [12]. We obtained 25% and 24% *O. volvulus* infectivity rate in black flies captured at Gurara and Kuratam breeding sites respectively as against zero prevalence reported by

Isiayaku et al [12]. Therefore, the high *O. volvulus* infectivity rate in black flies observed in this study indicated recrudescence of transmission of onchocerciasis in the study areas. There is possibility of imported infection from the neighbouring Niger State and FCT Abuja to Kaduna State. According to Crosskey [7], rivers Gurara, Kaduna, Sarkin Pawa (in Niger State) are always infested with black fly with rivers Gurara and Kaduna forming a major fly area of approximately 140 miles (225 km). The ability of *S. damnosum* complex to migrate >400 km away in large numbers [1,29,30] due to seasonal variation can threaten the interruption of transmission of onchocerciasis and elimination of the disease in Kaduna State.

To curb the black fly population density, slash and clear approach in Kaduna and the neighbouring endemic areas can be adopted. This approach requires removal of vegetation from breeding sites (fast flowing well oxygenated sediment free water) along rivers to disrupt *S. damnosum* breeding activities. Smith et al [31] recommended slash and clear using community-driven implementation strategy at least once a year for accelerating achievement of onchocerciasis elimination. Slash and clear vector control using community-driven implementation strategy at Maridi dam spillway in South Sudan reduced *S. damnosum* by >90% six months post intervention and <50% twelve months post intervention [32]. The approach has also been used in Uganda with 97% reduction in biting rate after single intervention during early wet season [33] and Cameroon with 32.9% reduction in black fly population density [34]. Combination of community directed treatment with ivermectin (CDTi) with slash and clear vector control will accelerate achievement of onchocerciasis elimination in Africa. The use of Esperanza Window Trap can also be deployed along black flies breeding sites to complement the slash and clear approach as described [35].

Despite the ongoing treatment with ivemectin for Lymphatic filariasis in Kachia and Kagarko LGAs after stopping treatment for onchocerciasis in 2018, the 0.13% Ov16 serological test result in children aged 5–9 years coupled with overall infectivity rate of 0.32 (C.I. 0.21-0.49) which is above the WHO threshold of <1/2000 infective rate in black flies indicates ongoing transmission of onchocerciasis in the study area. This implies that there is recrudescence of onchocerciasis in Kaduna State despite the interruption of transmission of onchocerciasis recorded in year 2018. The reasons for these observations may be attributed to insecurity, human migration and diagnostic technique used. The WHO onchocerciasis technical advisory subgroup in its sixth meeting [36,37] observed that real time qPCR assay used in this study has a higher sensitive than O-150 PCR ELISA and could detect residual transmission in fly pools negative for O-150 PCR ELISA. Furthermore, in a recent publication by Adeleke et al [35] on a comparative study of OvND5 qPCR and O-150 PCR ELISA in three ecological zones in Nigeria, they reported that OvND5 qPCR is more sensitive. More black fly pools were positive for *O. volvulus* by OvND5 qPCR compared with O-150 PCR in derived savannah (31.15 vs. 15.57%), montane forest (11.54 vs. 0%) and rainforest (23.08 vs. 2.56%).

Searching beyond the diagnostic technique used, the climate change with great impact on black flies' movement and human migration (intra and interstate) as a result of communal conflicts, political crisis, kidnapping and insurgency are also potential threats to elimination of onchocerciasis [38–40]. The earlier therapeutic coverage of 65% with ivermectin attained in Kaduna State before stopping MAM [12] might be insufficient to break the transmission cycle considering the high infectivity rate of black flies in some parts of the State.

Kagarko, Kachia and other LGAs in southern part of Kaduna State have had several episodes of insecurity since 2011 [41]. During the period of insecurity, the population in the affected communities moved out to other LGAs within the State and Niger State then returned when the situation was calm. The migration of people from affected communities poses a threat to elimination of onchocerciasis in other LGAs that share boundaries with the LGAs in Kaduna; Tafa and Suleja in Niger States and Bwari in FCT, Abuja. This calls for a review of geographic and therapeutic coverage rates, implementation of community self-monitoring and intensifying monitoring and evaluation of treatment. There is need for development of model that will determine appropriate frequency of treatment for Kaduna State. Adoption of biannual treatment in River Gambia, Senegal (April or May and October or November) for more than 10 years had been successfully implemented with desired outcomes [42]. Also, the biannual community directed treatment with ivermectin resulted to elimination of onchocerciasis in Colombia in 2013 [43]; Ecuador in 2014 [44] and Guatemala in 2016 [45]. To achieve elimination of onchocerciasis, Mexico changed frequency of treatment with ivermectin to quarterly and achieved elimination in year 2015 [46].

## Limitations of the study

The limitations of the study include inability to analyse all the 3107 DBS collected from children aged 5–9 years because of limited resources, devaluation of the naira, inflation and high cost of reagents during this study. Other limitations were failure to collect DBS from children in Amuse; a first-line community with the highest infectivity rate in black flies due to insecurity during the period of the study and inability to collect black flies in the dry season as a result of the short duration of the study.

## Conclusion

In conclusion, the high *O. volvulus* infectivity rate (above WHO threshold) in black flies and positive samples of DBS in children in some of the communities sampled showed that there is an ongoing transmission in the study area. This, by implication established the recrudescence of onchocerciasis after interruption of the disease was achieved in Kaduna in 2018. This calls for urgent programmatic evaluation of the extent of recrudescence since our study only covered two transmission foci and the key drivers of the recrudescence. Further entomological, epidemiological and social science studies are recommended to determine whether the recrudescence observed in this study is driven by vector migration, human migration, socio-cultural issues or insecurity.

## Supporting information

**S1 Methods. Detailed protocol for Ov16 ELISA.**
(S1_Methods.DOCX)

**S2 Methods. Detailed protocol for DNA isolation from black fly heads and qPCR.**
(S2_Methods.DOCX)

**S3 Methods. Detailed statistical variation in fly population density at various sites and across various months.**
(S3_Methods.DOCX)

**S1 Results. Detailed CT-values for pools of black flies examined.**
(S1_Results.XLSX)

## Acknowledgments

We acknowledge Sightsavers, Kaduna Office and Kaduna State Ministry of Health for availing us essential information during the design of the study protocols and sample collection. Also, we thank Nigerian Institute for Trypanosomiasis (and Onchocerciasis) Research, Kaduna for the provision of the vehicle used during field activities. Osun State University Multi-disciplinary Research Laboratory, Osogbo, Nigeria provided additional reagents for Ov16 serological test and qPCR analysis of *S. damnosum s.l.* Finally, we thank Mr. Samuel John Ayodele from Nigerian Institute for Trypanosomiasis (and Onchocerciasis) Research, Kaduna for producing the maps of the study areas and for his excellent technical support.

## Author contributions

**Conceptualization:** Timothy Oluwaseye Olanrewaju, Felicia N.C. Enwezor, Elizabeth O. Elhassan.

**Data curation:** Timothy Oluwaseye Olanrewaju, Felicia N.C. Enwezor, Oluwatosin B. Adekeye.

**Formal analysis:** Michael A. Igbe, Monsuru A. Adeleke.

**Funding acquisition:** Timothy Oluwaseye Olanrewaju, Felicia N.C. Enwezor, Luret A. Lar, Elizabeth O. Elhassan.

**Investigation:** Timothy Oluwaseye Olanrewaju, Felicia N.C. Enwezor, Michael A. Igbe, Ramatu A. Abdullahi.

**Methodology:** Timothy Oluwaseye Olanrewaju, Felicia N.C. Enwezor, Michael A. Igbe.

**Project administration:** Timothy Oluwaseye Olanrewaju, Felicia N.C. Enwezor.

**Supervision:** Luret A. Lar, Elizabeth O. Elhassan.

**Validation:** Felicia N.C. Enwezor, Luret A. Lar, Monsuru A. Adeleke, Oluwatosin B. Adekeye.

**Writing – original draft:** Timothy Oluwaseye Olanrewaju, Luret A. Lar, Michael A. Igbe.

**Writing – review & editing:** Timothy Oluwaseye Olanrewaju, Felicia N.C. Enwezor, Luret A. Lar, Michael A. Igbe, Ramatu A. Abdullahi, Monsuru A. Adeleke, Oluwatosin B. Adekeye, Elizabeth O. Elhassan.

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
