## [Decision Letter · Decision Letter 0]

14 Nov 2024

PNTD-D-24-01243Recrudescence of transmission of onchocerciasis in Kaduna State, Nigeria: What is the way forward?PLOS Neglected Tropical Diseases Dear Dr. OLANREWAJU, Thank you for submitting your manuscript to PLOS Neglected Tropical Diseases. After careful consideration, we feel that it has merit but does not fully meet PLOS Neglected Tropical Diseases's publication criteria as it currently stands. Therefore, we invite you to submit a revised version of the manuscript that addresses the points raised during the review process. Please submit your revised manuscript within 60 days Jan 13 2025 11:59PM. If you will need more time than this to complete your revisions, please reply to this message or contact the journal office at plosntds@plos.org.  Please include the following items when submitting your revised manuscript:* A rebuttal letter that responds to each point raised by the editor and reviewer(s). You should upload this letter as a separate file labeled '?>Response to ReviewersRevised Manuscript with Track ChangesManuscript

Shaden Kamhawi

co-Editor-in-Chief

Paul Brindley

co-Editor-in-Chief

**Additional Editor Comments (if provided):**
**Journal Requirements:**
**Reviewers' Comments:**

**Key Review Criteria Required for Acceptance?**

**Methods**

-Are the objectives of the study clearly articulated with a clear testable hypothesis stated?

-Is the study design appropriate to address the stated objectives?

-Is the population clearly described and appropriate for the hypothesis being tested?

-Is the sample size sufficient to ensure adequate power to address the hypothesis being tested?

-Were correct statistical analysis used to support conclusions?

-Are there concerns about ethical or regulatory requirements being met?

Reviewer #1: The objectives of the study were clear, and the design was fine. The population was clear and the sample collected: serological and entomological samples were sufficient.

Reviewer #2: The methods are satisfactory except that no statistical analysis was mentioned in the methods section

Reviewer #3: The objective of the study is clearly defined, articulated and appropriate for the study, however, no clear testable hypothesis was stated.T

The study area and population were well described, the study design is appropriate to address the stated objectives. The sample size was sufficient to ensure adequate statistical power to address the objectives, however, due to financial challenges encountered by the authors, not all samples collected were analyzed.

Consent and assent forms were signed for collection of dried blood spot, authors are been requested to indicate if vector collectors signed consent forms

**Results**

-Does the analysis presented match the analysis plan?

-Are the results clearly and completely presented?

-Are the figures (Tables, Images) of sufficient quality for clarity?

Reviewer #1: - Analysis presented match the analysis plan, although the serological samples analyzed were half of what WHO guidelines recommends, but serology alone is not enough for decision making. However, entomological samples and results obtained were adequate to determine that indeed disease recrudescence had occurred.

Reviewer #2: No method of data analysis was mentioned in the methods section

Data were mostly presented in raw form

Reviewer #3: The worth of data obtained from this study does not commensurate the results generated in this manuscript. A detailed statistical analysis of the data would have presented some more interesting results. Therefore, data analysis by the authors are not adequate for the kind of study carried out, unless the authors have already published some of the data elsewhere or have plans of publishing some of the data elsewhere.

Authors have beed asked to revise some tables and figures

**Conclusions**

-Are the conclusions supported by the data presented?

-Are the limitations of analysis clearly described?

-Do the authors discuss how these data can be helpful to advance our understanding of the topic under study?

-Is public health relevance addressed?

Reviewer #1: - the conclusions are supported by the data.

- limitations of analysis should be made clear.

- the authors have discussed how the data is important in understanding onchocerciasis recrudescence

- while discussing, the problem of public health, the authors should not that this is not only a Kaduna issue but also the neighbouring onchocerciasis endemic areas , and the human population force to move from place to place due to insecurity.

Reviewer #2: No clear conclusions were provided in the paper

Reviewer #3: Some of the conclusions are not well supported by the data presented. The authors did not effectively link some of their findings to the evidence they provided, this should be done.

Authors did not clearly state the limitations of the study, and has been asked to do so

The public health relevance of the study has been addressed

**Editorial and Data Presentation Modifications?**

Reviewer #1: No data modifications have been suggested

Reviewer #2: The article will require a thorough grammar check

Reviewer #3: This is an important study because it has demonstrated ongoing transmission in Kaduna State, where interruption of onchocerciasis transmission had been previously established in 2018. This posses a big threat to the onchocerciasis elimination targeted in 2030. This has been attributed to recent insecurity and the presence of migrants from other States where there could be ongoing transmission.

Nonetheless, the manuscript has a lot of technical, statistical and other challenges, and requires major revisions before it can be accepted for publication. In general, the manuscript is well written but have some grammatical mistakes and eros, these have been indicated in my review.

**Summary and General Comments**

Reviewer #1: This is a very important paper that should be published after suggested minor revision have been done.

Please see additional comments in the attachment

Reviewer #2: The article presents a topical matter of public health importance. However, the general presentation needs to be improved upon.

Tables 1 and 2 can be merged while retaining the substance contained therein

Discussion require close attention

I have made comments on the revised copy of the manuscript

Reviewer #3: General Comments

This is a comprehensive study, with the goal of determining recrudescence of onchocerciasis transmission in Kaduna State of Nigeria. The study employed entomological, serological and molecular biology techniques to determine blackfly infectivity in pooled blackfly heads using qPCR and collection of dried blood spot from human population to carry out OV16 AP-ELISA to assess seroprevalence in children aged 5-9 years in six of 23 local government areas (LGAs) in Kaduna State.

This is an important study because it has demonstrated ongoing transmission in Kaduna State, where interruption of onchocerciasis transmission had been previously established in 2018. This posses a big threat to the onchocerciasis elimination targeted in 2030.

The study has a lot of weakness, there are challenges with dried blood spot not collected from very relevant communities but no explaination provided, statistical analysis were not detailed. The authors should provide information on consent by vector collectors.

There is the need for majo revisions to be made, this has been capture in my reviewers comment.

PLOS authors have the option to publish the peer review history of their article (what does this mean? ). If published, this will include your full peer review and any attached files.

**Do you want your identity to be public for this peer review?** For information about this choice, including consent withdrawal, please see our Privacy Policy .

Reviewer #1: No

Reviewer #2: No

Reviewer #3: No

**Figure resubmission:****Reproducibility:** To enhance the reproducibility of your results, we recommend that authors of applicable studies deposit laboratory protocols in protocols.io, where a protocol can be assigned its own identifier (DOI) such that it can be cited independently in the future. Additionally, PLOS ONE offers an option to publish peer-reviewed clinical study protocols. Read more information on sharing protocols at https://plos.org/protocols?utm_medium=editorial-email&utm_source=authorletters&utm_campaign=protocols

---

## [Decision Letter · Decision Letter 1]

31 Mar 2025

PNTD-D-24-01243R1Recrudescence of transmission of onchocerciasis in some endemic communities in Kaduna State, Nigeria: What is the way forward?PLOS Neglected Tropical Diseases Dear Dr. Olanrewaju, Thank you for submitting your manuscript to PLOS Neglected Tropical Diseases. After careful consideration, we feel that it has merit but does not fully meet PLOS Neglected Tropical Diseases's publication criteria as it currently stands. Therefore, we invite you to submit a revised version of the manuscript that addresses the points raised during the review process. Please submit your revised manuscript within 30 days Apr 30 2025 11:59PM. If you will need more time than this to complete your revisions, please reply to this message or contact the journal office at plosntds@plos.org.  Please include the following items when submitting your revised manuscript: * A rebuttal letter that responds to each point raised by the editor and reviewer(s). You should upload this letter as a separate file labeled '?>Response to ReviewersRevised Manuscript with Track ChangesManuscript

Shaden Kamhawi

co-Editor-in-Chief

Paul Brindley

co-Editor-in-Chief

**Additional Editor Comments:**
**Journal Requirements:**

1) We note that your Old Manuscript file files are duplicated on your submission. Please remove any unnecessary or old files from your revision, and make sure that only those relevant to the current version of the manuscript are included.

2) Tables should not be uploaded as individual files. Please remove these files and include the Tables in your manuscript file as editable, cell-based objects. For more information about how to format tables, see our guidelines:

https://journals.plos.org/plosntds/s/tables 

**Reviewers' comments:**

**Key Review Criteria Required for Acceptance?**

**Methods:**

-Are the objectives of the study clearly articulated with a clear testable hypothesis stated?

-Is the study design appropriate to address the stated objectives?

-Is the population clearly described and appropriate for the hypothesis being tested?

-Is the sample size sufficient to ensure adequate power to address the hypothesis being tested?

-Were correct statistical analysis used to support conclusions?

-Are there concerns about ethical or regulatory requirements being met?

Reviewer #2: the methods are appropriate and suitable

Reviewer #3: The objective of the study is clearly defined, articulated and appropriate for the study, with clear testable hypothesis stated.

The study area and population were well described; the study design is appropriate to address the stated objectives. The sample size was sufficient to ensure adequate statistical power to address the objectives, however, due to financial challenges encountered by the authors, as a result not all samples collected were analyzed, and this has been indicated as a limitation of the study.

Consent and assent forms were signed for the collection of dried blood spots,

**Results:**

-Does the analysis presented match the analysis plan?

-Are the results clearly and completely presented?

-Are the figures (Tables, Images) of sufficient quality for clarity?

Reviewer #1: - The analysis presented did not match the analysis plan. A substantial number of serological samples was not analysed and biological transmission zones were never delineated.

- the results are clear, but the design does not indicate that they indicate disease recrudescence. It could just be failure of interventions to interrupt transmission.

- the Tables are good, but more clear and appropriate figures are required

Reviewer #2: yes

Reviewer #3: The authors have improved the statistical analysis to enhance the reporting of the results and discussion.

The authors have revised the tables and figures, and the results are clear and completely presented

Authors have clearly stated the limitations of the study.

**Conclusions:**

-Are the conclusions supported by the data presented?

-Are the limitations of analysis clearly described?

-Do the authors discuss how these data can be helpful to advance our understanding of the topic under study?

-Is public health relevance addressed?

Reviewer #2: yes

Reviewer #3: This is a comprehensive study, with the goal of determining recrudescence of onchocerciasis transmission in Kaduna State of Nigeria. The study employed entomological, serological and molecular biology techniques to determine blackfly infectivity in pooled blackfly heads using qPCR and collection of dried blood spot from human population to carry out OV16 AP-ELISA to assess seroprevalence in children aged 5-9 years in six of 23 local government areas (LGAs) in Kaduna State.

This is an important study because it has demonstrated ongoing transmission in some communities in the Kaduna State, where interruption of onchocerciasis transmission had been previously established in 2018. The current status of onchocerciasis in these areas poses a big threat to the onchocerciasis elimination targeted in 2030.

This study is of important public health significance and effort should be made to implement intervention strategies to eliminate onchocerciasis in these areas

The authors have clearly stated the limitations of the study.

**Editorial and Data Presentation Modifications?**

Reviewer #1: (No Response)

Reviewer #2: (No Response)

Reviewer #3: (No Response)

**Summary and General Comments:**

Reviewer #1: The study failed to explain the importance of delineating biological transmission zones in determining interruption of transmission of onchocerciasis or its recrudescence, and importance of appropriate maps. The paper could have been presented as “failure of a single annual dose of ivermectin to interrupt Transmission in Kaduna State in 29 years (1998-2018).”

Reviewer #2: Please reconcile information on lines 26-28 with that on lines 133-136

Provide a better map with higher resolution

Lines 176-178: what if you have more than one community within this distance, say one 8km and the other 17km?

Lines 249-250: Is it breeding sites that are washed away? please clarify

Lines 338-340: Further studies would determine whether the recrudescence observed in this study is driven by socio-cultural issues, human migration, fly migration or insecurity

Line 384: Chikezie

Reviewer #3: General Comments

This is a comprehensive study, with the goal of determining recrudescence of onchocerciasis transmission in Kaduna State of Nigeria. The study employed entomological, serological and molecular biology techniques to determine blackfly infectivity in pooled blackfly heads using qPCR and collection of dried blood spot from human population to carry out OV16 AP-ELISA to assess seroprevalence in children aged 5-9 years in six of 23 local government areas (LGAs) in Kaduna State.

This is an important study because it has demonstrated ongoing transmission in some communities in the Kaduna State, where interruption of onchocerciasis transmission had been previously established in 2018. The current status of onchocerciasis in these areas poses a big threat to the onchocerciasis elimination targeted in 2030.

This study is of important public health significance and effort should be made to implement intervention strategies to eliminate onchocerciasis in these areas

The authors have clearly stated the limitations of the study.

Following the revision of the manuscript by the authors, the manuscript has improved, however, some sections require minor revisions before it can be published.

The following corrections should be made to the manuscript:

Ln 28: The authors should indicate why the use of Diethylcarbamazine for human treatment was discontinued.

Ln 206: The statement, “the highest number 2,858 (71.48%) of black flies captured was from Kuratam community” is incorrect as should be revised.

Moreover, the total number of blackflies collected were 6,897, so the percentage of blackflies captured in Kuratam community cannot be 71.48% as indicated ; “2,858 (71.48%)”

Ln 308, the statement “returned when the situation is calm” should read “returned when the situation was calm”

The author has provided the supplement data on the qPCR analysis. However, the qPCR analysis showed significant numbers of O.ochengi infections in the blackfly population, but, the authors did not mention the presence of O. Ochengi in the study areas.

The authors should explain if the presence of O. Ochengi is important in the transmission dynamics of O. volvulus, and how this can influence the elimination of onchocerciasis in the study areas.

PLOS authors have the option to publish the peer review history of their article (what does this mean? ). If published, this will include your full peer review and any attached files.

**Do you want your identity to be public for this peer review?** For information about this choice, including consent withdrawal, please see our Privacy Policy .

Reviewer #1: No

Reviewer #2: No

Reviewer #3: No

**Figure resubmission:****Reproducibility:** To enhance the reproducibility of your results, we recommend that authors of applicable studies deposit laboratory protocols in protocols.io, where a protocol can be assigned its own identifier (DOI) such that it can be cited independently in the future. Additionally, PLOS ONE offers an option to publish peer-reviewed clinical study protocols. Read more information on sharing protocols at https://plos.org/protocols?utm_medium=editorial-email&utm_source=authorletters&utm_campaign=protocols

---

## [Editor Report · Decision Letter 2]

18 May 2025

PNTD-D-24-01243R2Recrudescence of transmission of onchocerciasis in some endemic communities in Kaduna State, Nigeria: What is the way forward?PLOS Neglected Tropical Diseases Dear Dr. Olanrewaju, Thank you for submitting your manuscript to PLOS Neglected Tropical Diseases. After careful consideration, we feel that it has merit but does not fully meet PLOS Neglected Tropical Diseases's publication criteria as it currently stands. Therefore, we invite you to submit a revised version of the manuscript that addresses the points raised during the review process. Please submit your revised manuscript within 30 days Jun 17 2025 11:59PM. If you will need more time than this to complete your revisions, please reply to this message or contact the journal office at plosntds@plos.org.  Please include the following items when submitting your revised manuscript: * A rebuttal letter that responds to each point raised by the editor and reviewer(s). You should upload this letter as a separate file labeled '?>Response to ReviewersRevised Manuscript with Track ChangesManuscript

We look forward to receiving your revised manuscript.

Shaden Kamhawi

co-Editor-in-Chief

Paul Brindley

co-Editor-in-Chief

**Additional Editor Comments:**

Recrudescence of transmission of onchocerciasis in Kaduna State, Nigeria: The way forward?

The manuscript describes an epidemiological study in Kaduna State, Nigeria conducted to detect possible recrudescence of onchocerciasis. The authors have responded to two rounds of reviews and the manuscript is much improved. There remain a few issues raised by the reviewers that the authors should account for before acceptance of the manuscript. Please see below.

Major

The authors did not include a version which accurately showed track changes. This makes it hard for reviewers to review properly. Please make sure documents are correct when submitting.

I believe it is PLOS NTD formatting rules to have the Methods before the Results section. Please check the style guide. The Limitations and Conclusions should stay after the Discussion.

The map is improved and useful, however, it is still small and hard to see. If the authors are concerned with the map blurring, consider 2 separate maps, one showing Chikun, and Kadunga North and South, and one showing Kachia and Kagarko. It was unclear where ZangonKataf was. I could not find it on the map.

As the rivers are referred to often in the manuscript, they should be labeled on the map. It would also help to have the symbols show which sites had positive DBS samples.

The authors make the key assertion that recrudescence has taken place in this area. However, one alternative explanation is that transmission was never actually interrupted in the first place. The authors cite the Tekle et al. and Isiyaku et al. papers as evidence of interruption, but don’t really interrogate the results in those papers. It would be very relevant to the community to use the Discussion section to compare the authors’ results (and the geographic areas studied) with the results in the earlier papers. Which data from those papers make the authors think that there was a definitive demonstration of interruption of transmission? Could some foci have been missed? Just repeating that Kaduna is a transmission zone is not enough; were there focal areas within Kaduna shown to have been interrupted? Were those areas near where this study took place? This discussion should help these authors make the case that this is recrudescence.

Minor

Abstract line 42, 0.32 should have a percentage (“) after it, correct? This same number does not have a percentage in the Results section either.

To non-Oncho specific readers it may be unclear as to the relevance of the UCL .371 in line 43. Consider adding both LCL and UCL, or explaining.

Line 56 suggest saying “…Kaduna State four years after a stop MAM decision.”

Line 74-77 can you provide citations for this sentence?

Line 77, after the word “include” remove the semicolon.

Line 78, impacts instead of impact.

Line 90 define MAM the first time it is used

Lin 90 space after the number 17

Table 1 is missing the prevalence number to go with the LCL and UCL.

Line 268 define DBS first time it is used.

Line 319 flies instead of fly

**Figure resubmission:****Reproducibility:** To enhance the reproducibility of your results, we recommend that authors of applicable studies deposit laboratory protocols in protocols.io, where a protocol can be assigned its own identifier (DOI) such that it can be cited independently in the future. Additionally, PLOS ONE offers an option to publish peer-reviewed clinical study protocols. Read more information on sharing protocols at https://plos.org/protocols?utm_medium=editorial-email&utm_source=authorletters&utm_campaign=protocols

---

## [Editor Report · Decision Letter 3]

7 Jul 2025

Dear Mr. Olanrewaju,

We are pleased to inform you that your manuscript 'Recrudescence of transmission of onchocerciasis in some endemic communities in Kaduna State, Nigeria:  What is the way forward?' has been provisionally accepted for publication in PLOS Neglected Tropical Diseases.

Best regards,

Scott D. Nash

Guest Editor

Jong-Yil Chai

Section Editor

Shaden Kamhawi

co-Editor-in-Chief

Paul Brindley

co-Editor-in-Chief

Thank you for your thorough responses to the reviewers' suggestions. There are no other suggestions at this time

---

## [Editor Report · Acceptance letter]

Dear Mr. Olanrewaju,

We are delighted to inform you that your manuscript, " 

Recrudescence of transmission of onchocerciasis in some endemic communities in Kaduna State, Nigeria:  What is the way forward?," has been formally accepted for publication in PLOS Neglected Tropical Diseases.

Best regards,

Shaden Kamhawi

co-Editor-in-Chief

Paul Brindley

co-Editor-in-Chief
